# Prevalence assessment of sexually transmitted infections among pregnant women visiting an antenatal care center of Nepal: Pilot of the World Health Organization's standard protocol for conducting STI prevalence surveys among pregnant women

Rubee Dev[1]*, Shambhu P. Adhikari[2], Anjana Dongol[3], Surendra K. Madhup[4], Pooja Pradhan[5], Sunila Shakya[6], Shrinkhala Shrestha[7], Sneha Maskey[8], Melanie M. Taylor[9,10]

1 Faculty of Nursing, University of Alberta, Edmonton, Alberta, Canada, 2 Department of Physiotherapy, Kathmandu University School of Medical Sciences, Dhulikhel, Kavre, Nepal, 3 Department of Obstetrics and Gynecology, Kathmandu University School of Medical Sciences, Dhulikhel, Kavre, Nepal, 4 Department of Microbiology, Kathmandu University School of Medical Sciences, Dhulikhel, Kavre, Nepal, 5 World Health Organization- Country Office, Lalitpur, Nepal, 6 Department of Obstetrics and Gynecology, Kathmandu University School of Medical Sciences, Dhulikhel, Kavre, Nepal, 7 Department of Community Programs, Kathmandu University School of Medical Sciences, Dhulikhel, Kavre, Nepal, 8 Department of Community Programs, Kathmandu University School of Medical Sciences, Dhulikhel, Kavre, Nepal, 9 Department of Global Programmes of HIV, Hepatitis, STI, World Health Organization, Geneva, Switzerland, 10 Centers for Disease Control and Prevention, Division of STD Prevention, Atlanta, Georgia, United States of America

* rubee@ualberta.ca

## Abstract

### Introduction

Sexually transmitted infections (STIs) are common during pregnancy and can result in adverse delivery and birth outcomes. The purpose of this study was to estimate the prevalence of STIs; *Chlamydia trachomatis* (CT), *Neisseria gonorrhoeae* (NG), *Treponema pallidum* (syphilis), *Trichomonas vaginalis* (trichomoniasis), and *Human Immunodeficiency Virus* (HIV) among pregnant women visiting an antenatal care center in Nepal.

### Materials and methods

We adapted and piloted the WHO standard protocol for conducting a prevalence survey of STIs among pregnant women visiting antenatal care center of Dhulikhel Hospital, Nepal. Patient recruitment, data collection, and specimen testing took place between November 2019-March 2020. First catch urine sample was collected from each eligible woman. Gen-eXpert platform was used for CT and NG testing. Wet-mount microscopy of urine sample was used for detection of trichomoniasis. Serological test for HIV was done by rapid and enzyme-linked immunosorbent assay tests. Serological test for syphilis was done using

**Data Availability Statement:** All data relevant to the study are included within the article. Raw data cannot be shared publicly because of the confidentiality issues as it includes the personal identifying information of the women. Deidentified data could be available from the Kathmandu University Institutional Data Access / Ethics Committee (contact via email at irc@kusms.edu. np) for researchers who meet the criteria for access to confidential data.

**Funding:** This study was made possible by support from the UNDP/UNFPA/UNICEF/WHO/World Bank Special Programme of Research, Development and Research Training in Human Reproduction (HRP), Department of Sexual and Reproductive Health and Research, World Health Organization, Geneva, Switzerland.

**Competing interests:** The authors have declared that no competing interests exist.

"nonspecific non-treponemal" and "specific treponemal" antibody tests. Tests for CT, NG and trichomoniasis were done as part of the prevalence study while tests for syphilis and HIV were done as part of the routine antenatal testing.

## Results

672 women were approached to participate in the study, out of which 591 (87.9%) met the eligibility criteria and consented to participate. The overall prevalence of any STIs was 8.6% (51/591, 95% CI: 6.3–10.8); 1.5% (95% CI: 0.5–2.5) for CT and 7.1% (95% CI: 5.0–9.2) for trichomoniasis infection. None of the samples tested positive for NG, HIV or syphilis. Prevalence of any STI was not significantly different among women, age $\leq$ 24 years (10%, 25/229) compared to women age $\geq$ 25 years (7.1%, 26/362) (p = 0.08).

## Conclusions

The prevalence of trichomoniasis among pregnant women in this sub-urban population of Nepal was high compared to few cases of CT and no cases of NG, syphilis, and HIV. The WHO standard protocol provided a valuable framework for conducting STI surveillance that can be adapted for other countries and populations.

## Introduction

Every day around the world, more than one million people acquire a sexually transmitted infection (STI) [1]. Some viral STIs, like human papillomavirus (HPV) and human immuno-deficiency virus (HIV), are still incurable and can be deadly, but some bacterial and protozoal STIs–like *Chlamydia trachomatis* (CT), *Neisseria gonorrhoeae* (NG), *Treponema pallidum* (syphilis) and *Trichomonas vaginalis* (trichomoniasis)–are curable and detectable with widely available but underutilized diagnostic platforms [2].

Most CT and NG infections in women are asymptomatic, but if left untreated can lead to serious complications, including pelvic inflammatory disease (PID), infertility, chronic pelvic pain and infant infection [3]. Women with trichomoniasis may be asymptomatic and may experience negative pregnancy outcomes such as preterm birth and an increased risk of PID among HIV-infected women [4]. Trichomoniasis has also been independently associated with increased risk for HIV acquisition and transmission [5]. Maternal HIV infection is associated with low-birth weight and increased susceptibility to infections in HIV-exposed uninfected infants [6]. Vertical transmission of syphilis (congenital syphilis) is the second leading global cause of stillbirth (malaria being the first) [7]. Screening for early detection followed by appropriate management of these infections could reduce the sequelae of complications and transmission.

To address this global and critical issue and to enable countries to reach targets set by the Sustainable Development Goals, the World Health Organization (WHO) developed the Global Health Sector Strategy on Sexually Transmitted Infections 2016–2021 [8]. The strategy positions the health sector response to STI epidemics as critical to the achievement of universal health coverage–one of the key health targets of the SDGs identified in the 2030 agenda for sustainable development. A key pillar of this strategy is STI surveillance. WHO recommends routine STI prevalence assessments among general populations of women and men to assess STI burden and guide programming [9]. To guide countries in implementing routine STI

prevalence surveys, WHO released a standard protocol to assess prevalence of STIs among pregnant women in 2018 [10]. Nepal was chosen as the first pilot site for implementation of this protocol.

WHO global estimates of STIs among general populations are based on STI prevalence assessments in the peer-reviewed literature. Unfortunately, few of such STI prevalence surveys are published each year and studies are even fewer from the South East Asia region. The limited availability of STI prevalence surveys results in limited representativeness of global estimates. To address this surveillance priority, WHO developed a standard protocol for STI prevalence surveys [10] to encourage countries, regions, and academic institutions to perform STI prevalence surveys to improve STI surveillance and program response at national, regional and global levels. Herein we describe the pilot of the WHO protocol and comment on the benefits of generating STI prevalence data for program response.

Information on the burden of STIs among general populations and among pregnant women in Nepal is limited. Available data are in the form of STI case reports which are known to drastically under-estimate STI burden due to limited access to STI diagnostic services. Various studies have been conducted to examine STI prevalence, mainly HIV and syphilis; however, most of them are conducted among high-risk populations such as sex workers (1.7% HIV and 3.9% syphilis), truck drivers (1.5% HIV and 5.3% syphilis), and male labor migrant workers (8% HIV and 22% syphilis) [11–13]. STI prevalence among such high-risk populations is reported to be high [9]. HIV is reported to be concentrated among key populations, with HIV prevalence at 5% among male sex-workers, 8.5% among transgender people, and 8.8% among males who inject drugs [14]. Information about assessment and trends in STI prevalence—whether, and by how much, the prevalence is increasing or decreasing, and which populations are affected—can help Nepal monitor its STI trends and provide information on the effectiveness of prevention and control measures.

This study aims to fill an existing gap in the collection of STI strategic information among pregnant women in Nepal. Studies to determine the prevalence of STIs can support program actions by providing information on population burden and supporting prevalence and incidence estimates. Hence, the main purpose of this study was to epidemiologically determine the prevalence of CT, NG, syphilis, trichomoniasis and HIV among pregnant women attending an antenatal care (ANC) center in Nepal by adapting the WHO standard protocol for STI prevalence surveillance.

## Materials and methods

### Study design

This was a prevalence survey of STIs among pregnant women. The survey was conducted between November 2019 –March 2020 at the antenatal care center of Kathmandu University Hospital (KUH)/Dhulikhel Hospital (DH), Kavre, Nepal. The Strengthening the Reporting of Observational Studies in Epidemiology (STROBE) checklist was utilized to ensure quality reporting during this cross-sectional study [15] (see S1 Checklist).

### Study site and population

Pregnant women attending antenatal care clinic at KUH/DH were enrolled in the study. DH is a not-for-profit community and university hospital that covers the rural, urban, and suburban population from Kavrepalanchowk, Sindhupalchowk, Dolakha, Sindhuli, Ramechhap, Bhaktapur and other surrounding districts. DH is also a tertiary care hospital where on an average, 60–70 pregnant women visit the antenatal clinic every day.

## Sample size and sampling technique

The sample size was determined based on the estimated prevalence of CT in Dhulikhel and prevalence of trichomoniasis in Kathmandu and surrounding geographical areas. CT infection was prevalent in 0.8% (95% CI: 0.5–1.5) of the women in a STI prevalence study conducted among married women in the rural Kavre district of Nepal [16]. Trichomoniasis was prevalent in 1.3% of the pregnant women attending Paropakar Maternity and Women's Hospital in urban Kathmandu [17]. As recommended for prevalence studies, we used n = $Z^2P(1-P)/d^2$ formula to calculate the sample size; where n is the sample size, Z is the statistic corresponding to 95% level of confidence, P is expected prevalence (obtained from similar studies), and d is precision (estimated not to exceed 1%) [18]. For a given level of observed CT and trichomoniasis prevalence, the minimum sample size needed for a survey was estimated to be 305 and 493 pregnant women. Using the given estimated sample size, we sampled 591 pregnant women for this prevalence survey.

We used convenience sampling technique to sample pregnant women. All women attending antenatal clinic at DH were approached by a trained research assistant (RA) who screened them for their eligibility and sequentially enrolled all eligible women who were interested to participate in the study.

## Eligibility criteria for the enrolment site and pregnant woman

ANC center at DH was purposively selected as: (i) it was the first entry point for all the pregnant women visiting for antenatal check-up; (ii) the site provided services to a sufficiently large number of clients; (iii) there was a presence of a reliable laboratory for processing of specimens and transport to the laboratory; (iv) there was an availability of treatment for infected pregnant women; (v) the site was accessible to surveillance staff; and (vi) their on-site staff members were willing to cooperate and were trained to conduct the survey.

Pregnant women were eligible for inclusion if they: (i) attended the clinic irrespective of their gestational age; (ii) were aged 18–49 years or were emancipated minors by marriage; and (iii) were candidates for routine laboratory examinations. Women were excluded if they had previously visited the clinic during the survey period to avoid duplication, had received treatment for any of the STIs within two weeks, and if they did not give informed consent to participate in the study.

## Study procedures

**Enrollment, consenting, counseling, and follow-up of participants.** The study team organized a one-day training on nature of the study, study protocol and study procedures for the RA and co-investigators (gynecologists, laboratory personnel, site coordinator) before initiation of the study. All team members received training on interviewing clients for determining eligibility, obtaining informed consent, sample collection, sample transportation, research confidentiality and participants' rights, and data collection and storage procedures.

Women attending ANC clinic were approached by an RA to screen for the eligibility and enroll in the study. Participating women were notified about the occurrence of the study and the opportunity to undergo tests for CT, NG and trichomoniasis as part of the prevalence study in addition to the routine antenatal testing done for syphilis and HIV. Women were informed that the results of syphilis and HIV tests will be extracted from the electronic health record (EHR) at the laboratory. A written informed consent was obtained from each woman. Minors emancipated by marriage were allowed to consent independently as adults.

Women were counseled on participation and testing for the STIs. Women received counseling and assurance that the results of testing would be confidential and would not be shared outside of the prevalence survey or clinic setting. Counseling was specifically on: (i) the

benefits of early detection and provision of treatment to prevent manifestations of untreated infection, (ii) potential risks associated with untreated STIs during pregnancy on both the mother and newborns; (iii) potential emotional stress of being diagnosed with a STI; (iv) the importance of treatment of sexual partners to avoid re-infection during pregnancy; and (v) the risk of disclosure to sexual partners of exposure to an STI. Participants with positive test results for any STIs were followed-up for the treatment. Three attempts were made to contact each patient. The follow-up required that the patients are confidentially notified of their positive test results and asked to return to the clinic for the treatment and post-test counseling. Women with negative test results were not followed-up and were asked to come to the clinic on their regular ANC visit.

**Laboratory tests.** Laboratory testing for CT, NG, and trichomoniasis was undertaken through collection of first catch urine sample, while serological testing for syphilis and HIV was performed on blood sample as per routine care at DH [4]. CT and NG tests were performed using GeneXpert kits donated to the hospital by Cepheid, USA (donation company); hence, the tests were performed free of cost for the study participants. In Nepal, the GeneXpert (Cepheid, USA) testing platform is available for tuberculosis testing but not for routine testing for CT/NG. As part of a collaborative and integrated program effort between the WHO country office in Nepal, the National Tuberculosis Center, and DH, the GeneXpert platform available at DH was used for all CT/NG tests in the current prevalence survey.

The same urine sample collected for CT and NG tests was also used for the test of trichomoniasis using wet mount preparation method of microscopic examination of motile protozoa. Serological testing for syphilis was done using non-treponemal antibody screening test i.e., Rapid Plasma Reagin (RPR- RFCL Diagnova Trepostat, Ranbaxy, Gurgaon, India), and specific treponemal antibody tests (Treponema pallidum haemagglutination- Fujirebio, Europe, N.V., Gent, Belgium) to confirm positive screening tests. Serological HIV test was done by AccuDiag™ HIV enzyme-linked immunosorbent assay (HIV- ELISA, Diagnostic Automation/Cotez Diagnostics, Inc., CA, USA) method that detects HIV antibodies and antigens in the blood. In case of a positive result, the ELISA test was followed by a Western Blot test to confirm the diagnosis. These tests were performed following the national algorithm of Nepal [4]. Specimens for serological testing of syphilis and HIV were collected as part of the routine ANC package (including hemoglobin, glucose, blood grouping, HBsAg, HIV and syphilis testing) at DH during the first ANC visit according to the hospital ANC protocol.

Samples obtained from pregnant women were collected and transported in an icebox to the microbiology laboratory of DH where the tests for CT, NG and trichomoniasis were performed. Most of the samples were analyzed on the same day of collection, and samples left for analysis were stored in a refrigerator at -20˚C until further analyses. The results of syphilis and HIV was extracted from the EHR. Test results were dispatched within a week. Women with positive test results were contacted by the RA using personal phone numbers and asked to return for treatment.

**Treatment.** Women with positive test results who came back to the clinic for follow-up were referred to the gynecologists, who explained the STI test result, discussed their concerns and the need for treatment, prescribed them medicine following national treatment guidelines [19], and discussed the need for partner testing. Women with positive test results for any of the tested STIs received treatment free of cost according to the study protocol.

## Statistical analysis

Descriptive analyses were used to describe the socio-demographic and reproductive characteristics of pregnant women. The results of continuous variables were expressed as the mean

(standard deviation [SD]), and the results of categorical variables were expressed as counts (percentages). The prevalence of overall and individual STIs were calculated and summarized as percent with 95% confidence intervals (CIs). To measure the overall prevalence of at least one of the five STIs, we defined a composite STI variable as "any STI", as a positive test for CT or NG or trichomoniasis or syphilis or HIV infection. Bivariate analyses were conducted to examine the association between the participant characteristics and CT and trichomoniasis infections, using Pearson's chi-square test ($\chi^2$ test) and the Fisher's Exact test where expected cell counts for exposure and outcome were less than five [20]. A p-value less than 0.05 was considered statistically significant. All analyses were done using Stata version 15 (Stata Corporation, College Station, TX, USA).

### Ethical approval

This study was approved by the World Health Organization Research Ethics Review Committee (WHO ERC.0003182), Nepal Health Research Council (NHRC 78/2019), and Kathmandu University School of Medical Sciences-Ethical Review Committee (KUSMS-IRC 37/19), Nepal. Women enrolled in the study provided written informed consent to participate. All methods were carried out in accordance with relevant guidelines and regulations.

## Results

### Socio-demographic characteristics

In total, 591 (87.9%) of 672 approached women met the eligibility criteria and were recruited in the study. The mean age of the women attending the clinic was 26 years (SD = 4.2). A majority (46.4%) of the women were primigravida. The mean age at marriage was 22 years (SD = 3.6) and 62.4% of the women reported to be either a housewife or unemployed. Majority (84.8%) of the women reported being tested for STIs in the past of which 84.9% reported having negative result for any of the STIs, while 15.1% did not know their test result (**Table 1**).

### Prevalence of STIs

The overall prevalence of any STIs among women attending antenatal clinic at DH was 8.6% (51/591, 95% CI: 6.3–10.8); with 1.5% prevalence of CT (9/591, 95% CI: 0.5–2.5), 7.1% (42/591, 95% CI: 5.0–9.2) trichomoniasis, and zero prevalence of NG, syphilis, and HIV respectively (**Table 2**). Trichomoniasis was the most commonly occurring infection. Only one woman (0.2%) had both CT and trichomoniasis infections. Prevalence of any STI was not significantly different among women, age $\leq$ 24 years (10%, 25/229) compared to women age $\geq$25 years (7.1%, 26/362) (p = 0.08). Prevalence of any STI was common among women who reported being a housewife (32 /369, 8.7%) and among the primigravida (24/274, 8.8%). However, none of the demographic variables collected were statistically associated with having CT or trichomoniasis (**Table 3**).

### Treatment of participants

A total of 42 women (82.4%) who tested positive for any of the STIs were called for follow-up. Of those, 19 women (2 CT and 17 trichomoniasis positive cases) came and received treatment, while 23 women refused to come for the treatment. Among women who refused, one woman had already delivered a baby, and the remaining either refused to come as they lived far from the hospital and did not want to return to clinic only to receive treatment or said they will come but did not come. Despite of multiple attempts to call, we could not follow-up with 9 women as their phones were unreachable.

**Table 1. Characteristics of pregnant women visiting antenatal care clinic in Dhulikhel Hospital, Nepal (N = 591).**

| | n (%) or mean (SD) |
|---|---|
| **Sociodemographic characteristics** | |
| Age (years) | 26 (4.2) |
| Age category (years) | |
| < 20 | 29 (4.9%) |
| 20–24 | 200 (33.8%) |
| 25–29 | 235 (39.8%) |
| 30–34 | 107 (18.1%) |
| ≥ 35 | 20 (3.4%) |
| Age of marriage | 22 (3.6) |
| Place of residence | |
| Urban | 332 (56.2%) |
| Rural | 259 (43.8%) |
| Education level of women | |
| No education | 21 (3.6%) |
| Primary | 234 (39.6%) |
| Some college or university | 202 (34.2%) |
| Completed university | 134 (22.7%) |
| Education level of husband | |
| No education | 17 (3.3%) |
| Primary | 207 (40.7%) |
| Some college or university | 170 (33.4%) |
| Completed university | 115 (22.6%) |
| Occupation of women | |
| Housewife | 369 (62.4%) |
| Self-employed | 100 (16.9%) |
| Salaried | 122 (20.6%) |
| Occupation of husband | |
| Unemployed | 20 (3.4%) |
| Self-employed | 268 (45.4%) |
| Salaried | 303 (51.3%) |
| **Reproductive characteristics** | |
| Gestational week | 25.6 (9.1) |
| Gravidity | |
| 1 | 274 (46.4%) |
| 2 | 203 (34.4%) |
| ≥ 3 | 114 (19.3%) |
| Number of live births | 1 (1) |
| Number of abortion/miscarriages | |
| No abortion/miscarriages | 471 (79.7%) |
| 1 | 99 (16.6%) |
| >1 | 21 (3.6%) |
| **STI test history** | |
| Ever tested for STIs in the past | |
| Yes | 501 (84.8%) |
| No | 90 (15.2%) |
| Past test result for STIs | |
| Unknown | 76 (15.1%) |

(*Continued*)

**Table 1.** (Continued)

|  | n (%) or mean (SD) |
|---|---|
| Negative | 425 (84.9%) |

N (number of complete observations), SD (standard deviation), STI (sexually transmitted infection)

## Discussion

This first pilot application of the WHO standard protocol for assessing the prevalence of CT and NG among pregnant women established a standard method for conducting STI prevalence surveys in Nepal. We identified a lower than expected prevalence of STIs among this group of pregnant women attending a sub-urban ANC center in Kavre, Nepal. The prevalence of trichomoniasis was comparatively higher among the women followed by CT.

*Trichomonas vaginalis* was identified in 7.1% of women using a non-sensitive method of microscopic detection and CT was prevalent among 1.5% of women using a GeneXpert platform. These prevalence values are higher compared to 5.4% prevalence of trichomoniasis and 0.8% prevalence of CT ascertained in a previous study among married women in rural Nepal [16], likely due to the fact that the study was conducted among non-pregnant women that may not have been sexually active. Other studies in high and low-middle income countries have found varying prevalence of TV in pregnant women [21–24]. Based on the low sensitivity of the TV wet mount, we expect the prevalence of trichomoniasis in our study population to be higher than what we report.

Though the prevalence of CT was low in this study, it is alarming as a recent studies have reported a significant association of CT with preterm birth and term preeclampsia mainly among the women aged <25 years [25, 26]. Several studies have also suggested the association between trichomoniasis and preterm delivery as well as low-birth weight infants [27, 28]. These findings demonstrate the importance of considering integration of routine screening and treatment of STIs during antenatal care in Nepal.

Despite the low prevalence of STIs among this population of pregnant women, these data provide valuable STI prevalence points for CT and trichomoniasis that can be used to generate estimates of prevalence and incidence of these infections using STI modeling tools [29]. The participation of the WHO country office in Nepal as well as the research collaborators at DH, formed a system and partnership that could be replicated for repeat STI prevalence surveys among pregnant women and other populations such as men. Further, the collaboration with the national center for tuberculosis program for the shared use of the GeneXpert platform for

**Table 2. Prevalence of STIs among pregnant women visiting antenatal care clinic in Dhulikhel Hospital, Nepal (N = 591).**

|  | Number positive | % (95% CI) |
|---|---|---|
| **Types of STI** |  |  |
| Chlamydia | 9 | 1.5 (0.5–2.5) |
| Gonorrhea | - | - |
| Syphilis | - | - |
| Trichomoniasis | 42 | 7.1 (5.0–9.2) |
| HIV | - | - |
| Any STI | 51 | 8.6 (6.3–10.8) |

CI (Confidence interval), HIV (Human Immunodeficiency Virus), STI (sexually transmitted infection)

**Table 3. Characteristics of study participants by types of STI (N = 591).**

| | Chlamydia (n = 9) | | Trichomoniasis (n = 42) | | Any STI (n = 51) | |
|---|---|---|---|---|---|---|
| | % (95% CI) | p-value | % (95% CI) | p-value | % (95% CI) | p-value |
| **Sociodemographic characteristics** | | | | | | |
| Age category (years)<br> <20<br> 20–24<br> 25–29<br> 30–34<br> ≥35 | -<br>55.6 (21.1–90.1) 22.2 (-6.6–51.1) 11.1 (-10.7–32.9) 11.1 (-10.7–32.9) | 0.312 | 4.8 (-1.8–11.3) 42.9 (27.7–58.0) 38.1 (23.2–52.9) 11.9 (1.9–21.8) 2.4 (-2.3–7.1) | 0.729 | 4.0 (-1.5–9.5) 46.0 (32.0–59.9) 36.0 (22.5–49.5) 10.0 (15.8–18.4) 4.0 (-1.5–9.5) | 0.296 |
| Age category (dichotomous)<br> ≤24 years<br> ≥25 years | 55.6 (21.1–90.1) 44.4 (9.9–78.9) | 0.297 | 47.6 (32.3–62.9) 52.4 (37.1–67.7) | 0.221 | 50.0 (35.9–64.0) 50.0 (35.9–64.0) | 0.088 |
| Place of residence<br> Urban<br> Rural | 33.3 (0.6–66.1) 66.6 (33.9–99.4) | 0.19 | 69.0 (54.9–83.2) 30.9 (40.6–48.9) | 0.081 | 62.0 (48.4–75.6) 38.0 (24.4–51.6) | 0.386 |
| Education level of women<br> No education<br> Primary<br> Some college/university<br> Completed university | 22.2 (-6.6–51.1) 44.4 (9.9–78.9) 11.1 (-10.7–32.9) 22.2 (-6.6–51.1) | 0.051 | 4.8 (-1.8–11.3) 33.3 (18.9–47.8) 38.1 (23.2–52.9) 23.8 (10.7–36.9) | 0.749 | 8.0 (0.4–15.6) 36.0 (22.5–49.5) 34.0 (20.7–47.3) 22.0 (10.4–33.6) | 0.351 |
| Occupation of women<br> Housewife<br> Self-employed<br> Salaried | 66.7 (33.9–99.4) 22.2 (-6.6–51.1) 11.1 (-10.7–32.9) | 0.798 | 61.9 (47.0–76.8) 26.2 (12.7–39.7) 11.9 (1.9–21.8) | 0.137 | 64.0 (50.5–77.5) 24.0 (12.0–35.9) 12.0 (2.9–21.1) | 0.164 |
| **Reproductive characteristic** | | | | | | |
| Gravidity<br> 1<br> 2<br> ≥3 | 22.2 (-6.6–51.1) 44.4 (9.9–78.9) 33.3 (6.0–66.1) | 0.308 | 52.4 (37.1–67.7) 35.7 (21.0–50.4) 11.9 (1.9–21.8) | 0.437 | 48.0 (33.9–62.0) 36.0 (22.5–49.5) 16.0 (5.7–26.3) | 0.826 |

N (number of complete observations), CI (confidence interval), CT (Chlamydia trachomatis), STI (sexually transmitted infection)

p-values were obtained using Chi-square or Fishers exact tests.

CT/NG along with tuberculosis demonstrated the availability and the importance of expanding the STI diagnostic capabilities of this and other multiplex testing systems.

Regardless of the stigma associated with STI testing, nearly all pregnant women who were approached participated in screening. However, the findings of this prevalence survey should be interpreted with the following limitations. First, this is not a national representative sample as data were only collected from one clinical setting and thus these results cannot be generalized for the entire country. Second, the zero prevalence of NG, HIV and syphilis suggests that increasing our sample size could have estimated more accurate prevalence of these infections by providing more accurate information on prevalence, narrowing margin of error, and identifying correlates of infection; however, resources and contextual factors for this pilot application limited our ability to increase the sample size. Future prevalence studies could determine the correlates of STIs using multivariate analysis and a larger sample size. In April 2020, the study was suspended due to the outbreak of SARs-CoV-2 in the country. On June 25, 2020,

due to the worsening COVID-19 epidemic, the ongoing closure of the outpatient ANC recruitment site, the reorganization of the laboratory services to support the COVID-19 response, and the obtainment of the required sample size needed, the study was stopped. It must be pointed out that the aim of this study was to enhance STI surveillance and not to provide clinical care. However, contacting women within a week (from sample collection to results dispatch) to refer them for treatment was valuable to our understanding of the challenges of STI treatment when patients must return for results. Further, refusal of more than half of the women to re-visit clinic just for the treatment of STI was alarming. Strategies to improve testing and point of care treatment on the same visit should be prioritized.

The global burden of curable STIs is high and has shown no decline based on the most recent WHO estimates [1]. National-level prioritization of STI control has fallen behind that of other infections such as HIV and hepatitis [30]. Better STI surveillance data can be used to guide clinical systems in enhancing STI testing services for pregnant women as well as other populations. Moving forward, further studies will be needed to explore the most cost-effective STI screening and treatment strategies among pregnant women at national and local levels.

## Conclusions

In this prevalence survey, we found an overall low prevalence of STIs among pregnant women attending an antenatal clinic in Kavre, Nepal. Trichomoniasis was the most commonly detected STI followed by CT. Adoption of WHO's STI surveillance protocol at the country level will help to examine the prevalence and trend of STIs among pregnant women and other populations and prioritize infection control strategies accordingly.

## Supporting information

**S1 Checklist. STROBE statement—Checklist of items that should be included in reports of *cross-sectional studies*.**
(DOC)

## Acknowledgments

The authors would like to express their gratitude to the Department of Obstetrics and Gynecology and the Department of Microbiology at the Dhulikhel Hospital, and all the participants of the study.

**Disclaimer:** The views expressed in this manuscript are those of the authors and do not necessarily represent the official position of the World Health Organization, the United States of America Centers for Disease Control and Prevention, or other affiliated organizations.

## Author Contributions

**Conceptualization:** Rubee Dev, Sunila Shakya, Melanie M. Taylor.

**Data curation:** Shambhu P. Adhikari, Sunila Shakya, Sneha Maskey.

**Formal analysis:** Rubee Dev.

**Funding acquisition:** Rubee Dev, Melanie M. Taylor.

**Investigation:** Rubee Dev, Anjana Dongol, Surendra K. Madhup, Sunila Shakya, Shrinkhala Shrestha, Sneha Maskey.

**Methodology:** Rubee Dev, Pooja Pradhan, Sunila Shakya, Melanie M. Taylor.

**Project administration:** Shambhu P. Adhikari, Anjana Dongol, Surendra K. Madhup, Pooja Pradhan, Sunila Shakya, Shrinkhala Shrestha, Sneha Maskey.

**Resources:** Anjana Dongol, Surendra K. Madhup, Pooja Pradhan, Shrinkhala Shrestha.

**Software:** Sneha Maskey.

**Supervision:** Shambhu P. Adhikari, Anjana Dongol, Pooja Pradhan, Sunila Shakya, Melanie M. Taylor.

**Validation:** Anjana Dongol, Surendra K. Madhup, Shrinkhala Shrestha.

**Visualization:** Surendra K. Madhup.

**Writing – original draft:** Rubee Dev, Melanie M. Taylor.

**Writing – review & editing:** Rubee Dev, Shambhu P. Adhikari, Anjana Dongol, Sunila Shakya, Shrinkhala Shrestha, Melanie M. Taylor.

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
