## [Decision Letter · Decision Letter 0]

2 Mar 2021

PONE-D-21-02130

Prevalence assessment of sexually transmitted infections among pregnant women visiting an antenatal care center of Nepal: Pilot of the World Health Organization’s standard protocol for conducting STI prevalence surveys among pregnant women

PLOS ONE

Dear Dr. Dev,

Thank you for submitting your manuscript to PLOS ONE. After careful consideration, we feel that it has merit but does not fully meet PLOS ONE’s publication criteria as it currently stands. Therefore, we invite you to submit a revised version of the manuscript that addresses the points raised during the review process.

We look forward to receiving your revised manuscript.

Kind regards,

R Matthew Chico, MPH, PhD

Academic Editor

PLOS ONE

Journal Requirements:

3. Please upload a copy of Supplementary File which you refer to in your text on page 7.

Reviewers' comments:

Reviewer's Responses to Questions

**Comments to the Author**

1. Is the manuscript technically sound, and do the data support the conclusions?

Reviewer #1: Yes

Reviewer #2: Yes

2. Has the statistical analysis been performed appropriately and rigorously? 

Reviewer #1: Yes

Reviewer #2: No

3. Have the authors made all data underlying the findings in their manuscript fully available?

Reviewer #1: Yes

Reviewer #2: Yes

4. Is the manuscript presented in an intelligible fashion and written in standard English?

Reviewer #1: Yes

Reviewer #2: Yes

5. Review Comments to the Author

Reviewer #1: The Dev et al. manuscript describes a cross-sectional study of STI prevalence in Nepal. The authors report a relatively high prevalence of trichomoniasis (TV) compared to other STI. The work presented in this manuscript is well described and provides a valuable contribution to global health literature.

I am pleased to see the sound logically structure and concise writing style found in this manuscript, along with the solid methodology. I have a few questions and a few minor comments.

Questions:

1. Would the authors please clarify why the sampling is purposive as opposed to convenience? My understanding is that the authors did not target a specific subset of the ANC population in order to develop a specific sample population profile or have a specific goal needing a specific population, but instead sampled a general ANC population, so this seems to be convenience sampling?

2. As the authors acknowledged, the sensitivity of wet mount for TV (averaging around 50%) and PCR for CT (98.7%) is different. Although PCR for CT and TV is similar, sensitivity of wet mount for TV can be as low as 38%. The STI prevalence data in this manuscript is valuable, but I think the comparison might be problematic given the difference in sensitivity of the diagnostic tests. Perhaps decreasing the language that suggests a comparison might be more accurate?

3. The number of women in the sample that report STI testing history is reported as 84.7%. This seems high? And makes me wonder if there is something unique about this population?

Minor comments:

Line 92-93: Is there a specific reason that Nepal was chosen as the first pilot site for implementation of the new WHO protocol?

Line 201: This seems to be a repeat of line 199?

Line 211: I don’t think Wet should be capitalized?

Line 212 and Line 214: I don’t think the nonspecific/specific and the quotation marks around non-treponemal and treponemal are needed?

Line 221: HBsAg not HBsAG

Line 300 to 303: I am unsure about this argument as the prevalence of STIs is generally higher in Brazil than Nepal?

Reviewer #2: This is an important manuscript about the prevalence of STIs including HIV in pregnant women in Nepal. There are some areas of the manuscript that could be improved upon prior to publication listed below.

1. Abstract: introduction should include harmful effects on pregnancy outcomes and infants. Can you add that the syphilis and HIV testing were part of standard of care, but other STIs were part of the study (otherwise it appears that 12% of the sample were not tested for HIV/RPR).

2. Introduction: Line 75 "may be asymptomatic OR may experience" - should this be and as these are not mutually exclusive categories (symptoms and outcomes).

- It would be helpful to add in the prevalence reported in other studies in the introduction even in high risk populations or the study of married women in Nepal. Also please cite the HIV prevalence from UNAIDS for Nepal to give context to the study.

3. Methods: were partners informed about the partner's STI diagnosis? Were they referred for treatment or testing? how long did it take to test and contact participants?

For statistical analysis, it may be better to conduct a mutlivariate analysis that controls for other covariates like age, gestational age, education, etc if you want to test hypotheses of associations between STI diagnosis and age, education, etc. using a multivariate logistic regression model. Otherwise, I would avoid making hypotheses and presenting statistics from statistical tests as presented.

4. Results:

Table 1. Did you record what kind of STI tests women had had in past as the % seems high. Was that for HIV or syphilis in prior pregnancies? And why do the # differ in ever testing and unknown STI test results (90 vs 89 and 501 vs 502)?

Please include abortion/miscarriage in the line "no abortion" as they were combined in the question.

The analysis in table 3 presents simple chi-square stats and the p-value is almost significant for education and CT and TV and place of residence, and age and any STI (can update this with the %s in line 269-271). If the authors could conduct a multivariate analysis testing some of these hypotheses, the results would be more interesting/valid.

5. Discussion: please include what (if anything) was done for male partners? How long did it take to give feedback on the results? Why did only 1/2 of women return for treatment? What can be done to improve this (point of care testing, earlier feedback, partner engagement?).

Please add this outcome about limited treatment uptake to the abstract and conclusions/recommendations if there is space.

6. PLOS authors have the option to publish the peer review history of their article (what does this mean?). If published, this will include your full peer review and any attached files.

Reviewer #1: No

Reviewer #2: No

---

## [Author Response · Author response to Decision Letter 0]

21 Mar 2021

Manuscript reference number: PONE-D-21-02130

Title: Prevalence assessment of sexually transmitted infections among pregnant women visiting an antenatal care center of Nepal: Pilot of the World Health Organization’s standard protocol for conducting STI prevalence surveys among pregnant women

Dear Editor/Reviewers,

Thank you for your careful review of our manuscript entitled “Prevalence assessment of sexually transmitted infections among pregnant women visiting an antenatal care center of Nepal: Pilot of the World Health Organization’s standard protocol for conducting STI prevalence surveys among pregnant women” (PONE-D-21-02130).

We greatly appreciate your comments and have revised the manuscript and have detailed responses to each of the points below. 

Journal Requirements

We carefully checked the PLOS ONE’s style requirements and have revised our manuscript accordingly.

Thank you for this suggestion. In our revised cover letter, we have addressed the suggested points related to the data availability. 

3. Please upload a copy of Supplementary File which you refer to in your text on page 7.

The Supplementary File that we are referring to is STROBE Research Checklist. Revised version of the checklist now has been uploaded.

Thank you for this suggestion. We have now omitted the ethics statement from the declaration section and added it under the Methods section on page 13 of the revised manuscript.

Reviewer 1 

Questions

1. Would the authors please clarify why the sampling is purposive as opposed to convenience? My understanding is that the authors did not target a specific subset of the ANC population in order to develop a specific sample population profile or have a specific goal needing a specific population, but instead sampled a general ANC population, so this seems to be convenience sampling?

We would like to thank the reviewer for pointing this out. We agree with the reviewer’s comment and have now replaced the sampling technique from “purposive” to “convenience” on page 8, line 174 of the revised manuscript.

2. As the authors acknowledged, the sensitivity of wet mount for TV (averaging around 50%) and PCR for CT (98.7%) is different. Although PCR for CT and TV is similar, sensitivity of wet mount for TV can be as low as 38%. The STI prevalence data in this manuscript is valuable, but I think the comparison might be problematic given the difference in sensitivity of the diagnostic tests. Perhaps decreasing the language that suggests a comparison might be more accurate?

Thank you for the comment. We agree with this description regarding the large difference in sensitivity of the tests. 

In response, we have removed this statement in the discussion: A study conducted in Brazil reported similar prevalence of trichomoniasis (7.7%) among young pregnant women using a highly sensitive nucleic acid amplification test [21], supporting the sensitivity of detection in our study.

We have replaced the statement on page 17 with: Other studies in high and low-middle income countries have found varying prevalence of TV in pregnant women (21-24). Based on the low sensitivity of the TV wet mount, we expect the prevalence of trichomoniasis in our study population to be higher than what we report. 

3. The number of women in the sample that report STI testing history is reported as 84.7%. This seems high? And makes me wonder if there is something unique about this population?

Thank you for this comment. As part of the antenatal care in Nepal, every pregnant woman has to undergo general screening for STIs including HIV and syphilis during their first visit to the antenatal care clinic. However, the screening does not include a test for CT, NG, and trichomoniasis. More than half of the women (53.7%) in the sample were either 2nd or 3rd gravida, which indicates they might have undergone screening tests during their initial pregnancies. Even for the prima gravida, we recruited women, irrespective of their gestational age, which means most of them might already have undergone general screening for STIs. This could be the reason behind the high percentage of women reporting STI testing history.

Minor comments

Line 92-93: Is there a specific reason that Nepal was chosen as the first pilot site for implementation of the new WHO protocol?

Thank you for this comment. Nepal was selected as the first pilot site for implementation of the new WHO protocol because of the following reasons: (i) information on the burden of STIs among pregnant women in Nepal and in South East Asia was lacking, (ii) we were able to identify a setting with well-established laboratory setting that was already using GeneXpert platform for screening tuberculosis, and (iii) Two of the on-site researchers/clinicians involved in this study had conducted a similar study among married non-pregnant women in Nepal (Shakya et al., 2018). We believed that our proposed study would benefit from their experiences and expertise in conducting a similar study.

Reference:

Shakya S, Thingulstad S, Syversen U, Nordbø SA, Madhup S, Vaidya K, et al. Prevalence of Sexually Transmitted Infections among Married Women in Rural Nepal. Infectious diseases in obstetrics and gynecology. 2018;2018.

Line 201: This seems to be a repeat of line 199?

Thank you for pointing this out. We have now deleted the repetitive line from the manuscript.

Line 211: I don’t think Wet should be capitalized?

We have now replaced “Wet” with “wet”.

Line 212 and Line 214: I don’t think the nonspecific/specific and the quotation marks around non-treponemal and treponemal are needed?

We have now omitted the quotation marks.

Line 221: HBsAg not HBsAG

We have now replaced “HBsAG” with “HBsAg”.

Line 300 to 303: I am unsure about this argument as the prevalence of STIs is generally higher in Brazil than Nepal?

We agree with the reviewer’s comment regarding the higher prevalence of STIs in Brazil compared to Nepal. 

Based on this comment we have edited the following statements in the discussion as noted above:

We have removed this statement in the discussion: A study conducted in Brazil reported similar prevalence of trichomoniasis (7.7%) among young pregnant women using a highly sensitive nucleic acid amplification test [21], supporting the sensitivity of detection in our study.

We have replaced the statement on page 17 with: Other studies in high and low-middle income countries have found varying prevalence of TV in pregnant women (21-24). Based on the low sensitivity of the TV wet mount, we expect the prevalence of trichomoniasis in our study population to be higher than what we report. 

Reviewer 2 

1. Abstract: introduction should include harmful effects on pregnancy outcomes and infants. Can you add that the syphilis and HIV testing were part of standard of care, but other STIs were part of the study (otherwise it appears that 12% of the sample were not tested for HIV/RPR).

Thank you for this suggestion. We have now paraphrased the introduction section of the abstract to specify harmful effects of STIs on pregnancy outcomes and infants. As suggested, we have also added the following sentence under the methods section on page 3, line 48-50 of the revised manuscript:

“Tests for CT, NG and trichomoniasis were done as part of the prevalence study while tests for syphilis and HIV were done as part of the routine antenatal testing.”

2. Introduction: Line 75 "may be asymptomatic OR may experience" - should this be and as these are not mutually exclusive categories (symptoms and outcomes).

Thank you for pointing this out. We agree with the reviewer’s comment regarding the categories not being mutually exclusive. We have now replaced ‘or’ with ‘and’.

- It would be helpful to add in the prevalence reported in other studies in the introduction even in high-risk populations or the study of married women in Nepal. Also please cite the HIV prevalence from UNAIDS for Nepal to give context to the study.

Thank you for this suggestion. We have now added the prevalence of STIs for the high-risk populations. We have also added the HIV prevalence from UNAIDS for Nepal and cited the given below reference:

Reference:

UNAIDS. Country Progress Report Nepal. To contribute to global AIDS monitoring report 2017. Accessed on 15 March 2021 at https://www.unaids.org/sites/default/files/country/documents/NPL_2018_countryreport.pdf.

3. Methods: were partners informed about the partner's STI diagnosis? Were they referred for treatment or testing? how long did it take to test and contact participants?

Thank you for these questions. The study team did not inform about the women’s STI diagnosis to their partners. Only women were informed about their diagnosis who were then called for a follow-up counseling and treatment. During the counseling, the gynecologists also discussed about the need for partner testing with women. We have specified this under the treatment section. Test results were ready within a week. Women with positive test results were then contacted by RA by calling in their personal phone numbers to come to the clinic for a follow-up visit for the treatment. We have added the following sentence on page 12, line 261-262:

“Test results were dispatched within a week. Women with positive test results were contacted by the RA using personal phone numbers and asked to return for treatment.”

For statistical analysis, it may be better to conduct a multivariate analysis that controls for other covariates like age, gestational age, education, etc. if you want to test hypotheses of associations between STI diagnosis and age, education, etc. using a multivariate logistic regression model. Otherwise, I would avoid making hypotheses and presenting statistics from statistical tests as presented.

We understand the reviewer’s concern regarding the conduct of multivariate analysis controlling for other covariates; however, it was beyond the scope of our study. To address the issue, we have omitted the given below paragraph from the manuscript:

The high rate of CT and trichomoniasis infection in this study was observed among younger women ≤29 years (86%, 95% CI: 76.3-95.7), which is similar to the findings of a study conducted among pregnant women in Egypt where the high rate of infection was reported in age group of 20–30 years (29). Younger women are at higher risk of acquiring infection because of a combination of behavioral, biological, and cultural reasons. For some STDs, such as CT, young women may have increased susceptibility to infection because of increased cervical ectopy, which although is a normal finding in young women, make them more susceptible to infection (30). This study further highlights the need to strengthen the efforts to screen and treat STIs during antenatal care, mainly among the younger age group women.

Further, we added the given below statement under the limitations section:

“Future prevalence studies could determine the correlates of STIs using multivariate analysis and a larger sample size.”

4. Results:

Table 1. Did you record what kind of STI tests women had had in past as the % seems high. Was that for HIV or syphilis in prior pregnancies? 

The investigators were not able to record what kind of STI tests women had had in the past, as the women did not remember the name of the test nor did they have any record of the test. It was completely based on women’s memory of the past that might have led to the high percentage of STI test reporting. Women did not specify whether the test was done for HIV or syphilis in prior pregnancies as they were not informed, thus, highlighting the lack of proper counseling during the antenatal care.

And why do the # differ in ever testing and unknown STI test results (90 vs 89 and 501 vs 502)?

We would like to thank the reviewer for pointing this issue out. Past test result for STIs should have been among the women who had ever tested for STIs in the past, i.e., 501. We have now updated the numbers in Table 1.

Please include abortion/miscarriage in the line "no abortion" as they were combined in the question.

As suggested, we have now included abortion/miscarriages in the line “no abortion” of Table 1.

The analysis in table 3 presents simple chi-square stats and the p-value is almost significant for education and CT and TV and place of residence, and age and any STI (can update this with the %s in line 269-271). If the authors could conduct a multivariate analysis testing some of these hypotheses, the results would be more interesting/valid.

In this study, a p-value less than 0.05 was considered statistically significant, hence, we did not include the borderline significance (p=0.08) as statistically significant result. We have added the following statement under the statistical analysis section on page 13, line 293-294:

“A p-value less than 0.05 was considered statistically significant.”

While we understand the reviewer’s concern regarding the conduct of multivariate analysis, it was beyond the scope of our study. Future prevalence studies could determine the correlates of STIs using multivariate analysis.

We have added the following acknowledgement of this limitation in the discussion section:

“Future prevalence studies could determine the correlates of STIs using multivariate analysis and a larger sample size.”

5. Discussion: please include what (if anything) was done for male partners? How long did it take to give feedback on the results? Why did only 1/2 of women return for treatment? What can be done to improve this (point of care testing, earlier feedback, partner engagement?).

Please add this outcome about limited treatment uptake to the abstract and conclusions/recommendations if there is space.

Thank you for this valuable suggestion. Due to the stigma associated with STIs, male partners were not directly informed about the women’s STI results. Women were informed about their result directly over a phone call and asked to come for the treatment. Women who came back for the treatment were then referred to the gynecologists who along with the treatment discussed about the need of partner testing with the women. This information has been included under the treatment section of the manuscript.

For the duration it took to give feedback on the results, we added “contacting women within a week (from sample collection to results dispatch)” on page 19, line 430-431.

Regarding the refusal of treatment and what can be done to improve it, we have included the following sentence under the discussion section on page 19, line 432-434:

“Further, refusal of more than half of the women to re-visit clinic just for the treatment of STI was alarming. Strategies to improve testing and point of care treatment on the same visit should be prioritized.”

Since the main aim of the pilot study was to enhance STI surveillance and not to provide clinical care, the authors did not highlight about the limited treatment uptake in the abstract.

The thoughtful comments and guidance of the PLOS ONE reviewer is greatly appreciated. The authors feel that the reviewers’ revisions as well as other small clarifications have strengthened the focus and content of this manuscript submission. We look forward to the PLOS ONE decision.

Kind regards,

Rubee Dev, PhD, MPH

University of Alberta

Faculty of Nursing

Edmonton, Canada

Email: rubee@ualberta.ca

Phone: +1 778-821-2375

---

## [Editor Report · Decision Letter 1]

6 Apr 2021

Prevalence assessment of sexually transmitted infections among pregnant women visiting an antenatal care center of Nepal: Pilot of the World Health Organization’s standard protocol for conducting STI prevalence surveys among pregnant women

PONE-D-21-02130R1

Dear Dr. Dev,

We’re pleased to inform you that your manuscript has been judged scientifically suitable for publication and will be formally accepted for publication once it meets all outstanding technical requirements.

Kind regards,

R Matthew Chico, MPH, PhD

Academic Editor

PLOS ONE
---

## [Editor Report · Acceptance letter]

14 Apr 2021

PONE-D-21-02130R1 

Prevalence assessment of sexually transmitted infections among pregnant women visiting an antenatal care center of Nepal: Pilot of the World Health Organization’s standard protocol for conducting STI prevalence surveys among pregnant women 

Dear Dr. Dev:

I'm pleased to inform you that your manuscript has been deemed suitable for publication in PLOS ONE. Congratulations! Your manuscript is now with our production department. 

Kind regards, 

on behalf of

Dr. R Matthew Chico 

Academic Editor

PLOS ONE